# Efficient Informative Path Planning via Normalized Utility in Unknown Environments Exploration

**DOI:** 10.3390/s22218429

**Published:** 2022-11-02

**Authors:** Tianyou Yu, Baosong Deng, Jianjun Gui, Xiaozhou Zhu, Wen Yao

**Affiliations:** Defense Innovation Institute, Chinese Academy of Military Science, Beijing 100071, China

**Keywords:** informative path planning, exploration, autonomous robot, navigation

## Abstract

Exploration is an important aspect of autonomous robotics, whether it is for target searching, rescue missions, or reconnaissance in an unknown environment. In this paper, we propose a solution to efficiently explore the unknown environment by unmanned aerial vehicles (UAV). Innovatively, a topological road map is incrementally built based on Rapidly-exploring Random Tree (RRT) and maintained along with the whole exploration process. The topological structure can provide a set of waypoints for searching an optimal informative path. To evaluate the path, we consider the information measurement based on prior map uncertainty and the distance cost of the path, and formulate a normalized utility to describe information-richness along the path. The informative path is determined in every period by a local planner, and the robot executes the planned path to collect measurements of the unknown environment and restructure a map. The proposed framework and its composed modules are verified in two 3-D environments, which exhibit better performance in improving the exploration efficiency than other methods.

## 1. Introduction

Recently, autonomous robots have begun to be used to replace human work [1,2,3,4], even in harsh environments, such as battlefields, caves, and extraterrestrial environments [5,6]. In such scenarios, communication is infrequent or limited, manual operation is difficult for persistently collecting environmental data. A robot’s perception of the unknown environment and independent planning ability in such scenarios is particularly important [7].

The process of robot autonomous movement and environment map building is called unknown environment exploration [8]. Using a Micro Aerial Vehicle (MAV) to explore in an unstructured environment is common research. Due to its high degree of motion flexibility, it is able to complete the motion track with high maneuver requirements [9]. A MAV equipped with computing units, vision, and positioning sensors can collect the information measurements to perceive and map the environment in real-time. MAV moves independently without prior information on the global environment but the real-time map is based on records from an airborne sensor.

If the environment is completely unknown in advance, it is difficult to formulate a globally optimal solution to control the MAV by a series of inputs at one time. The most common method is receding horizon control [10,11,12], which iteratively determines a control input to navigate the robot to scan unknown space. For the navigation that leads to information measurements, a seminal method is detecting the frontier, which is identified as the boundary between the known and unknown regions of the map [13,14]. Even if a robot greedily seeks the frontiers, it indicates a series of feasible exploring actions. However, it is so simple that it lacks a comprehensive evaluation of the candidate region. The robot works without considering the information gathered, some decisions that execute an information-less path will cause reverse and high path distance costs [15]. And the frontiers detection process is time-consuming in a largely high-dimensional environment map, so it is difficult to guarantee real-time performance in local planning [16]. The sample-based method is commonly used for high-dimensional planning efficiency. It samples the candidate viewpoint and raw path and considers the utility along each candidate path. Information measurement and path cost are considered in the utility calculation to determine an informative path. An informative path which navigates the robot to information-rich areas can improve the efficiency of exploration [17].

In this article, we propose an efficient exploration solution based on informative path planning via normalized utility. A road map is extended by generating viewpoints randomly, it is maintained through the whole exploring process to utilize prior information in path decisions. Information measurements at the candidate viewpoints are pre-calculated to judge the potential unknown space volume, the posterior map entropy decline value is used to evaluate the information-richness of the measurements. A utility normalizing the information measurements by the path cost is proposed for deciding an optimal informative path. For a smooth motion, a minimum snap [18] trajectory is generated from the waypoints set that map out the optimal path. Octomap [19] is used to map the environment and divide the space into occupied, free, and unknown states. Overall, our contributions mainly lie in three aspects:An Efficient information richness judgment from posterior map entropy decline value is proposed, which formulates the potential unknown detection volume for navigating robots to visit unknown space efficiently.An informative path utility calculation method that normalizes information measurements by path distance is proposed; the normalized utility leads to fewer local optimums.The proposed method has been extensively validated in two realistic simulation environments.

The rest of this article is organized as follows. In Section 2, we introduce related work in autonomous robotic exploration. A Problem Statement is proposed in Section 3. In Section 4, we present our proposed method. Experiments and results are given in Section 5. We conclude our work and provide future directions in Section 6.

## 2. Releated Work

Autonomous exploration is the main prerequisite for robots to build a map of unknown environments to provide relevant data. Although there are many techniques for efficient and autonomous exploration in recent research, it is still challenging to decide on an informative path for measurement collection.

The major approaches can be grouped into the frontier [20,21,22,23] and sample-based [24,25,26] methods. The boundary between observed and unobserved space is called the frontier. Frontier-based exploration computes the frontiers periodically to maintain a candidate goal set and navigates the robot to a goal that optimally trades off the path cost and frontiers cover. Detecting frontiers as a goal guarantees full coverage of the area [13], but the detection calculation is time-expensive for travel space voxels in a map of large environments. To relieve the time-consumption of detection in [8], frontiers outside of the FOV (field of View) are a secondary priority to decrease computation and maintain faster motion. Similarly, searching the region for frontier detection is bound in [15]; however, the advantage of coverage guarantee is weakened, and the reverse becomes general. Another disadvantage of the frontier-based method is its lack of consideration of information measurements. Navigation following the frontiers indeed guarantees full coverage but ignores information-richness. Even if a robot wanders greedily, in an information-richness space, it can still collect many information measurements. Sample-based approaches sample viewpoints, paths, or control inputs, e.g., RRT [27,28,29,30,31], PRM (Probabilistic Roadmap) [32,33,34]. The optimal goal is chosen by comparing the utility estimation of each sampled candidate. The utility formation permits a broad range of utility objectives to adopting different kinds of task requirements. In [10], Bircher et al. use the exponential utility to decide a next-best-view path; unknown volume in the FOV at the viewpoint is considered to be an award, an exponent tuning factor is used to penalize high path costs. However, the behavior of exploration is sensitive to the size of the factor; too large or too small a value will result in repetitive exploration trajectories [27]. In [9], the frontiers are considered to be awards. Eungchang et al. use the exponential utility in path decision and active loop-closing planning, a likelihood-based selection is developed to reduce the drift of pose estimation, and motion velocity is considered to prevent inappropriate loop-closing, which may cause repetitive motion. In [35,36], the linear utility trade-off between path cost and frontiers award linearly, but it needs to adjust the parameters carefully to avoid inefficient back-traveling [37]. To relieve the repetition path problem, a well-designed utility is supposed to be modeled. In this paper, as a trial, we employ a normalized utility, for which we use the notion of efficiency, i.e., the accumulated gain per cost, as a central idea for the value.

The above research neglects information uncertainty of some voxel when only considering the frontiers or the unknown volume to be the award, and it will also result in repetitive exploring. Driving the robot to information-rich areas can prevent repetitive paths, and help to escape from the local optimum. For an information-richness-oriented exploration strategy, recent research thus oftentimes supports a sample-based local planner with additional global information to improve coverage [38,39]. Several methods have been proposed that maximize the information-gathered volume. This predicts a decreasing volume of uncertainty by future sensor measurements. For example, in [40], the unknown volume and quality of the reconstructed surfaces are both considered to lead an efficient exploration. In [27], a global planner considers the frontiers to be the global goals, and a local planner uses sample-based method combined with the evaluation by information gain. To evaluate the information gain more efficiently, in [27,41], the Gaussian process is used to build a continuous and differentiable gain space. Charrow et al. [17] consider a sample-based method as a local exploring planner, the frontier as a global planner to make up the local minimum disadvantage, and use Cauchy–Schwarz quadratic mutual information (MI), which is more computing-efficient. However, these methods evaluate information gain in the whole grid environment map [15]; although they optimize more accurately, it is time expensive to search over all the 3-dimensional maps. In this paper, we calculate the information measurements by only evaluating the voxels in the FOV at the viewpoint. The Shannon entropy is calculated to evaluate the uncertainty.

In summary, in this paper, we focus on the planning, and use the sample-based method for its efficiency and maintain an incremental topological road map, formulate the information measurement of each sampled viewpoint by calculating the decreasing map entropy. The raw path is sought in the roadmap, guaranteeing the fast informative path decision. To avoid the local minimum, a normalized utility is used, it calculates average information on each unit path distance.

## 3. Problem Statement

The task of UAV exploration in an unknown environment performs the process of exploring and mapping iteratively. A 3D workspace W of known size is given before the task for establishing the concerned area; all UAVs will explore the workspace. Exploration processing by identical UAVs with four degrees of freedom, at the 3D position ${\left[x,y,z\right]}^{T}\in R^{3}$ and the yaw angle $\psi \in S^{1}$. The UAV state can be described as $\xi ={\left[x,y,z,\psi \right]}^{T}$. In each platform, a depth camera is equipped to collect the environment information with a certain field of view.

A 3-D occupancy grid mapping is run for reconstructing a volumetric map M of the environment. The occupancy probability of each gird m∈M is initialized as P(m)=0.5. The posterior occupancy probability $P\left(m|\xi_{1:t},z_{1:t}\right)$ is updated by the depth measurement $z_{1:t}$ and the UAV state $\xi_{1:t}$ from initial time to current time *t*. The grids in the map will be gradually scanned by the sensor and identified as either free grids $M_{f}=\{m|P\left(m|\xi_{1:t},z_{1:t}\right)<P_{free}$, m∈M} or occupied grids $M_{o}=\{m|P\left(m|\xi_{1:t},z_{1:t}\right)>P_{occ}$, m∈M}. $P_{f}$ and $P_{o}$ are the given thresholds.

Given a map $M_{t}$ at time *t*, its map uncertainty can be denoted by entropy [42]:(1)$$\begin{array}{c} H\left(M_{t}\right)=\sum_{m\in M_{t}}-p\left(m\right)\log_{2}p\left(m\right)-\left(1-p(m)\right)\log_{2}\left(1-p(m)\right). \end{array}$$

The predicted information measurement $I\left(M_{t};\xi_{i},z_{i}\right)$ [42] at state $\xi_{i}$ is formulated as:(2)$$I\left(M_{t};\xi_{i},z_{i}\right)=H\left(M_{t}\right)-H\left(M_{t}|\xi_{i},z_{i}\right).$$

It can be used to quantify the information measurement at a waypoint $p_{i}=\xi_{i}$ and evaluate the path P, which is determined by key waypoints. In the informative path planning process, the receding horizon exploration planner decides an optimal path $P^{*}$ in every period. To seek the $P^{*}$ for the UAV so that it gathers measurements that reduce unknown space with less consumption, a cost function is formulated to measure the value of the candidate path, considering the uncertainty of map M, the location of waypoints in path P, and the time cost of the path c(P).
(3)$$P^{*}=argmax_{P}\;f\left(M_{t},P,c\left(P\right)\right)=argmax_{P}\;\frac{I(P)}{c(P)},\;s.t.\;c\left(P\right)\leq B,$$
where *B* denotes a time budget. I(P) defines the collected information measurements along the path P, and is defined in more detail in (Equation 11).

UAVs visit unknown spaces independently according to the outputs of the exploration planner. We assume that the UAVs are equipped with an accurate localization system. The core parts of our proposed modules are as follows.

## 4. Method

### 4.1. System Overview

The system overview of the proposed autonomous exploration framework is shown in Figure 1. The Depth sensor is a depth camera with a FOV of ${\left[60,\;90\right]}^{\circ }$, which is equipped on the UAV. The localization in this paper is assumed to be perfect and can provide real-time odometry of the UAV. The 3-D occupancy grid-mapping thread is run to build the model of the environment during the exploration, providing the information of interest. A sample-based exploration local planner builds a topological road map via RRT incrementally; the road map provides a candidate goal set. Periodically, the planner determines an optimal informative path via normalized utility; the utility is considered to trade off the information measurements and path cost along the navigation route.

### 4.2. Mapping

A numerically environmental expression is necessary for exploring. In a unified space, the environment could be divided into the unknown, occupied, or free parts. Due to its simple and fast searching character, the OctoMap [19] is adopted in our method.

If the up-to-date sensor measurements $z_{1:t}$ are given, the probability updating of a voxel $L\left(v|z_{1:t}\right)$ can be formulated:(4)$$L\left(v|Z_{1:t}\right)=\max\left(\min\left(L\left(n|Z_{1:t-1}\right)+L\left(n|Z_{t}\right),l_{\max}\right),l_{\min}\right),L\left(v\right)=\log\left[\frac{P(v)}{1-P(v)}\right].$$

After every depth measurement is received, a ray-casting operation is used to update the occupancy probabilities of voxels along the beams. The probability map provides the uncertainty information and guarantees the calculation basis for navigation.
(5)$$L\left(v|Z_{t}\right)=\left\{\begin{array}{cc} l_{\mathrm{occ}}=0.85 & \;\mathrm{if}\;\mathrm{reflected}\\ l_{\mathrm{free}}=-0.4 & \;\mathrm{if}\;\mathrm{traversed} \end{array}\right..$$

### 4.3. Topological Road Map

To build a road map $T=[n_{0},n_{1},...,n_{N}]$, the node $n_{i}=\left[n_{parent},n_{child},\xi_{n_{i}},I\left(M_{t};\xi_{n_{i}},z_{n_{i}}\right)\right]$, $\xi_{n_{i}}$ is the robot state of the $n_{i}$, $n_{parent}$ is the parent node of $n_{i}$, and $n_{child}$ is the child node of $n_{i}$. Initially, the first node is initialized as $n_{0}=\left[\emptyset ,\emptyset ,\xi_{init},I\left(M_{t};\xi_{init},z_{init}\right)\right]$, while $\xi_{init}$ represents the start state of the robot for the exploration.

According to Algorithm 1, until the termination of the exploration, the road map is maintained by extending new nodes which are generated from random sampling. The candidate is randomly placed in W. The FindNearest(C,T) finds a nearest node of C in 3-dimensional euclidean space:(6)$$n_{near}=\arg min_{n\in T}∥{\left[I_{3},0\right]}_{4\times 3}\xi_{n}-C∥.$$
**Algorithm 1** Road Map Extension1:T=[∅].2:**for Exploration** is not over **do**3:   $C\leftarrow X∼Uniform\left(W\right),X\in R^{3}$.4:   $n_{near}\leftarrow \mathrm{FindNearest}\left(C,T\right)$.5:   $success\leftarrow \mathrm{CollsionFree}(C,n_{near})$.6:   **if** success
**then**7:     ξ←(C,BestYaw(C)).8:     $z\leftarrow \mathrm{Predict}(M_{t},\xi )$.9:     $n_{new}\leftarrow \left[n_{near},\emptyset ,\xi ,I\left(M_{t};\xi ,z\right)\right]$.10:     $n_{near}\leftarrow \left[n_{parent},n_{new},\xi_{n_{near}},I\left(M_{t};\xi_{n_{near}},z_{n_{near}}\right)\right]$11:     $T\leftarrow \mathrm{Extend}(n_{new},T)$.12:   **end if**13:**end for**

Then, collision detection is supposed to be done by connecting them in 3-dimensional euclidean space, as Figure 2 shows, the blue point shows the position of the candidate, and the blue dotted line indicates that the collision detection result is free. If it is collision-free, then BestYaw is used to get the best yaw to scan unknown space. The future state ξ is fixed and $n_{new}$ is defined and extended to the road map.

The BestYaw uses a method based on the gradient of the weight function, which is inspired from [34], and can be computed as:(7)$$n_{\mathrm{view}}=\sum_{c\in N}w^{\prime }\left(c\right)\frac{c-x}{∥c-x∥},x={\left[I_{3},0\right]}_{4\times 3}\xi .$$
(8)$$w^{\prime }\left(c\right)=\left\{\begin{array}{cc} 1 & \;\mathrm{if}\;\mathrm{voxel}\;c\;\mathrm{is}\;\mathrm{unknown},\;\\ 0 & \;\mathrm{otherwise}.\; \end{array}\right.$$
where $N=\sum_{v\in M\cap A}v$ is the set of the voxel in an area delineated by a circle, x is the center, and the perception range is the radius. A new sensor configuration is generated along the direction $n_{\mathrm{view}}$ at point x, and it can be denoted as a state ξ. The unknown voxels mentioned later are of the highest information uncertainty, which means that more information may be observed in the corresponding space. And otherwise, space states no longer have high uncertainty because they have reached the threshold of occupancy probability. Two intuitive examples are given in Figure 3.

To navigate, the robot arrives at the location with rich information and the information measurement $I(M_{t};\xi ,z)$ is used to formulate the information-richness. It is calculated after *z* is predicted according to the current map $M_{t}$ and ξ. As the Figure 2 shows, when the ξ is fixed, the map voxel $m\in M_{z}=FOV\cap M_{t}$ can be confirmed. According to (Equation 2):(9)$$\begin{array}{cc} I\left(M;\xi ,z\right)= & H\left(M_{z},M_{z}^{-}\right)-H\left(M_{z},M_{z}^{-}|\xi ,z\right)\\ = & H\left(M_{z}\right)+H\left(M_{z}^{-}\right)-H\left(M_{z}|\xi ,z\right)-H\left(M_{z}^{-}|\xi ,z\right)\\ = & H\left(M_{z}\right)\\ ≃ & -\sum_{m\in FOV\cap M_{t}}p\left(m\right)\log\left(p\left(m\right)\right)+\left(1-p\left(m\right)\right)\log\left(p\left(m\right)\right). \end{array}$$

We make $H\left(M_{z}|\xi ,z\right)=0$, and $H\left(M_{z}^{-}\right)=H\left(M_{z}^{-}|\xi ,z\right)$, as we assume the robot can perfectly decrease the uncertainty of $M_{z}$ when it is at state ξ, the voxels in the FOV will be fully known with future observation. The voxels out of the FOV will not be scanned and the posterior occupancy probability will not change.

As the (Equation 9) shows, the ideal decrease of map uncertainty can be seen as a sum of binary entropy. For one voxel, its occupancy probability increases from 0 to 1. It becomes flat when the logarithm base increases; and when probability increases, the entropy reaches the maximum at 0.5. This means that when a voxel *v* is unknown, v=0.5 and it is in the most uncertain state, from the perspective of information measurement, we can scan more information here.

### 4.4. Informative Path Decision

In the proposed informative path planner, an optimal informative path is decided in every period. Both the information measurements and path cost are considered to judge how worthy to execute is a given candidate path; in other words, how information-rich the path. The path decision process can be denoted by (Equation 3). And $f(M_{t},P,c\left(P\right))$ can be formulated as a kind of normalized utility:(10)$$f\left(M_{t},P,c\left(P\right)\right)=\frac{I(P)}{c(P)},$$
specifically:(11)$$P^{*}=\arg max_{P}\;\frac{I(P)}{c(P)}=\arg max_{P}\;\frac{\sum_{i=1}^{k}I\left(M_{t};p_{i},z_{i}\right)}{\sum_{i=0}^{k-1}c\left(p_{i+1},p_{i}\right)}.$$

The path $P=\left[p_{0},p_{1},...,p_{k}\right],k\in Z^{+},p_{i}\in T$, is the sum of segments $<p_{j},p_{j+1}>$, where j=0,1,2,...,k−1. $<p_{j},p_{j+1}>$ denotes a raw collision-free path between node $p_{j}$ and $p_{j+1}$. P is the variable in the optimization process, while *k* is not a constant number, it changes with the node corresponding to the end of the candidate path. The $z_{i}$ corresponds to observation at state $\xi_{p_{i}}$. The $c(p_{i+1},p_{i})$ denotes the distance between $p_{i}$ and $p_{i+1}$ in 3-dimensional euclidean space:(12)$$c\left(p_{i+1},p_{i}\right)=∥{\left[I_{3},0\right]}_{4\times 3}\xi_{p_{i+1}}-{\left[I_{3},0\right]}_{4\times 3}\xi_{p_{i}}∥.$$
An intuitive example can be given by Figure 4, and $c\left(P^{*}\right)=\sum_{i=0}^{k-1}c\left(p_{i+1},p_{i}\right)$ is equal to the total length of green line segment. The k=5 in $P^{*}$ in first row, k=6 for the second row.

There are also other common formations of utility, e.g., Exponential [27,40] and Linear [36]:(13)$$f\left(M_{t},P,c\left(P\right)\right)=\left\{\begin{array}{cc} \sum_{i=1}^{k}I\left(M;p_{i},z_{i}\right)-\lambda \sum_{i=1}^{k-1}∥p_{i+1}-p_{i}∥ & \;\mathrm{Linear}.\;\\ \sum_{i=1}^{k}I\left(M;p_{i},z_{i}\right)\cdot e^{-\lambda ∥p_{i}-p_{0}∥} & \;\mathrm{Exponential}. \end{array}\right.$$

The Linear method linearly combines information measurements and path cost. The Exponential method formulates the utility of the next step as a Markov iteration process; $I\left(M;p_{i},z_{i}\right)$ along the path will be multiplied by an exponential attenuation term decreasing with increasing distance. Both of them are sensitive to λ, with bigger or smaller λ. The robot, using Linear, will reverse very frequently. For Exponential, the exploring behavior becomes very limited to the current region, and more likely to ignore available unknown space located far away from the current location. The normalized utility calculates average information on each unit path distance; it has very intuitive physical significance and avoids the parameter problem.

The $I(M_{t};p_{i},z_{i})$ is restored in T, the T is maintained and updated constantly. The $P^{*}$ is decided from the current T, see Figure 4. Each node n∈T will be traveled and a candidate path that starts from the current state to the $\xi_{n}$ along the segments will be calculated to judge how information-rich it is. The best path with maximal utility will be the decided informative path. To realize a receding horizon control, the first segment $<p_{0},p_{1}>$ will be executed; see the blue segment in Figure 4, the green best path corresponds to the predicted result in a period, and the blue is the motion control input.

### 4.5. Continuous Trajectory

Given the raw path $P^{*}$, continuous trajectories are required for smooth navigation. Our UAV trajectory planning is based on a method [18] that generates smooth and dynamically feasible trajectories. The trajectory is essentially a high-order polynomial spline, all parameters of the high-order splines are optimized, so that the total trajectory time is minimized to enable the quadrotor to fully utilize its dynamic capability.

These high-order splines are generally used for local trajectory generation and have many advantages, including the ability to specify velocities, accelerations, and lower derivatives at waypoints, very fast evaluation times, and compact representation of long and complex trajectories. While a closed-form solution exists to minimize the sum of squared derivatives of such a spline, the concerned freedom of motion contains four degrees ξ=x,y,z,ϕ; it can be considered as only planned outputs in a reduced space of differentially flat outputs [43].

## 5. Experiments

In order to verify the efficacy and efficiency of the proposed method, extensive experiments using the Gazebo simulation engine are conducted, in which the experimental environment and simulation robot share the same physical properties as that of the real world [44]. The software system is implemented on ROS Melodic release on top of an Ubuntu version 18.04LTS operating system, with a laptop with Intel Core i7-12700H CPU at 2.6 GHz, 32-GB memory.

Exploration is evaluated in an indoor flat environment of size $20\times 12\times 3\;m^{3}$ and a maze scenario of $15\times 15\times 2\;m^{3}$(Figure 5). The planner parameters in both environments are given in (Table 1). The mapping process, exploration completion, and total completion time are recorded to evaluate the methods.

### 5.1. Effect of Normalized Utility

We first compare our normalized utility with both the Exponential and Linear in two scenarios. Except for different formations, all other settings are the same. Every planner has been run 20 times in two scenarios with the same initial position.

Exploring processes at three common time points of algorithms are given to show the dynamic process in indoor exploring (Figure 6). The exploring completion degree curves are also given in (Figure 7). As Figure 6 shows, at 300s, the Exponential and Normalized complete the exploration, but the Linear only completes about 80%. The trajectory is smooth, but frequent local backtracking appears in the Exponential, especially in the lower right corner, see Figure 6b, the frequent reversing makes a bottleneck of exploration. In Figure 7, the area from about 50s to 150s reflects the global backtracking, that the robot travels along a long path without new information. From about 185s to 230s reflects the local backtracking that the robot frequently reverses in a short distance. As for Linear, it traps at about 80% completion. See Figure 6a where the robot is trapped in repeated reversing more than 300s, and always performs a breadth-first behavior. The normalized method is 12 s faster than the Exponential. The Exponential is faster in the first 60s, but falls into the local minimum, the Normalized keeps a high exploring speed till about 75s, and starts backtracking to scan the last unknown space in a far distance. The Normalized is better both in mean and standard deviation of completion, see Figure 7b, the Linear always fails to complete the exploration so it is not depicted. The exploring completion rate of each stage is shown more clearly in Figure 8. Obviously, Linear performs at a lower rate in the whole process. The Exponential falls into a platform period earlier from 65s to 150s, and the corresponding percentage completed is nearly equal to zero. Although the Normalized falls into the platform period later, it performs similarly to the Exponential.

The proposed Normalized method is more efficient because it is not easy to get stuck in the local minimum, the Exponential always focuses on the local area and fails to detect tasks far away, and the Linear prefers a breadth-first search and may fail a local reverse. When exploring, some voxels are unknown and counted in a utility calculation but cannot be scanned as occlusion, imperfect perception of the depth sensor, or failure of map update. Especially, the occlusion will cause the local reverse, the voxels that cannot be updated immediately make some corresponding viewpoints maintain valuable for information measurements.

The maze is explored with the same configuration. It is more challenging due to the narrow corridors, closed dead corners, and looping space. The same experimental method is carried out and the same key data are recorded. The Normalized performs better in this scenario. As Figure 9 shows, both the Linear and Exponential fail to complete the scanning of the whole maze when the Normalized finished. The Linear performs breadth-first search behavior and frequently backtracks through the maze. The Exponential falls into the local minimum in the narrow gap in the lower left corner of the map; see Figure 9b. Repeated wandering corresponds to the exploring bottleneck that lasted for a long time; see Figure 10a, the red curve from about 185s to 230s. Not surprisingly, the Exponential is the fastest at the beginning until 150s, while the Normalized maintains a persistently high rate until 200s, and completes it about 90s ahead of the Exponential. It can be seen clearly in Figure 11.

The maze is more challenging, and the algorithm that focuses on local scanning may fall into the local minimum and waste time. As Figure 10b shows, the minimum completion time difference between Normalized and Exponential is small (about 15s); but on the whole, its standard deviation and mean are bigger, and the performance gap is larger than in the indoor scenario.

### 5.2. Viewpoint Evaluation

To validate the efficiency of the proposed viewpoint-information-richness-evaluation method, the frontier-based evaluation method [13] is compared in the above two different simulation environments. The frontiers in the FOV are counted for evaluating the viewpoint. All comparative experiments use the Normalized method for path utility calculation. Except that the evaluation process is different, the other configuration is the same.

The results are shown in Table 2. The frontier calculation in a limited FOV consumes less than the information uncertainty calculation. In the above two scenarios, the evaluation computation time of a viewpoint is less than 1ms, while the information uncertainty computation time is about 1∼5ms. However, the exploration using frontier to guide consumes more. In the indoor scenario, the total time is 280.2±30.3s for frontier, and it only consumes 220.5±21.8s for ours. In the maze scenario, the total time is 330.3±42.1s, and it only consumes 225.8±35.2s for ours.

Our information-based navigation is better than the frontier-based method in overall efficiency, although it costs more time in one evaluation. Actually, the inefficient navigation to the unknown-less space is more time-expensive. Figure 12 shows the process of exploring using frontier and our information. Especially, the difference in the maze is quite obvious. Using frontier as a judgment of unknown space cannot guarantee the information content around the goal state; it navigates the robot to a local minimum sometimes due to the rarefied information content around the goal. As Figure 12a shows, from about 100s to 210s, the exploring rate suddenly descends compared with the rate before 100s. Also, in the maze, as Figure 12b shows, the robot falls into a stage of low efficiency from 100s to 210s. The information uncertainty calculation in FOV guides the robot to a space that is unknown-rich. This guarantees efficiency in exploring, and this advantage is more prominent in the maze. A suboptimal decision is more likely to be determined in this challenging scenario, especially, when a frontier evaluation method is used.

## 6. Conclusions

In this paper, a novel informative path-planning method is proposed to realize the unknown exploration. To validate the efficiency of the proposed method and the system, extensive experiments were conducted. The proposed solution uses an RRT based method to incrementally build and maintain a topological road map and evaluates the path by normalized utility considering the information-richness and the traveling cost, which improves the exploring efficiency in general. The road map efficiently provides us with an initial raw path for receding horizon control. Overall, the proposed method performs better than other common approaches.

In the future, it is suggested that the utility of the coverage of candidate nodes should be considered, as a path that guarantees a certain coverage guides the robot to visit more valuable nodes at once, and so may reduce the backtracks in the later stage of exploration. The information-driven method should be improved by considering the probabilistic correlation, which may enhance efficiency and decreases the time consumption of evaluation. The frontier-based method should be considered to ensure coverage in the later stage.

## Figures and Tables

**Figure 1 sensors-22-08429-f001:**
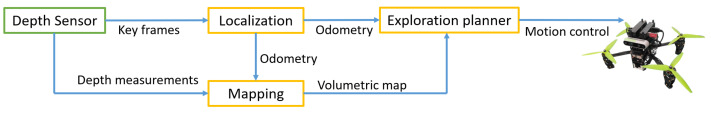
The system overview. The modules of localization, mapping, and planner are run on the UAV. The UAV visits unknown spaces independently according to the outputs of the exploration planner.

**Figure 2 sensors-22-08429-f002:**
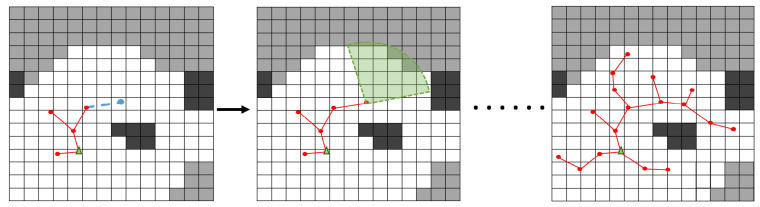
Roadmap is built as a living RRT. We keep the tree alive and maintain it through the whole exploration process. The green triangle represents the current state of the robot, while the green sector represents the FOV. The red lines represent the maintained roadmap, and the red dots represent the nodes. The blue point shows the position of the candidate, and the blue dotted line indicates that the collision detection result is free. The white, gray, and black grids represent free, unknown, and occupied space respectively.

**Figure 3 sensors-22-08429-f003:**
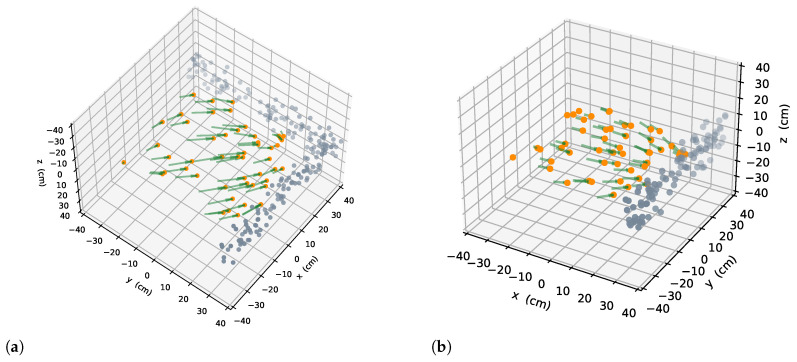
Example of best yaw in two scenarios, each gray point depicts the position of an unknown voxel; a green arrow with an orange dot shows the best scanning direction. (**a**) Scenario one. (**b**) Scenario two.

**Figure 4 sensors-22-08429-f004:**
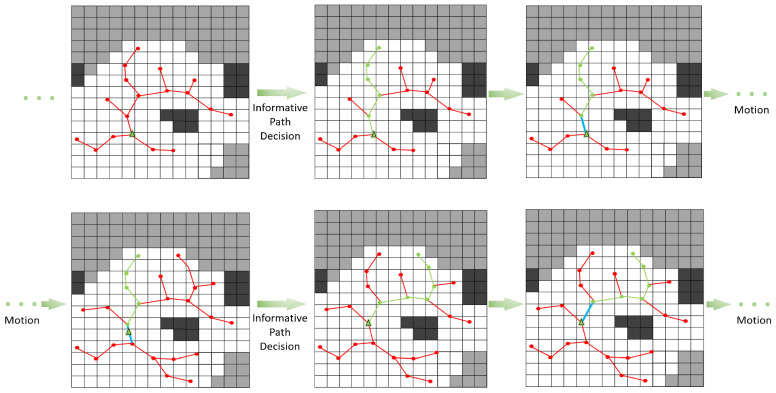
Process the planning loops till the termination of the exploration. The green triangle represents the current state of the robot, while the green lines represent the optimal informative path and the blue line represents the executive path in one period. The red lines depict the maintained roadmap. The red and green dots represent the nodes. The white, gray, and black grids represent free, unknown, and occupied space respectively.

**Figure 5 sensors-22-08429-f005:**
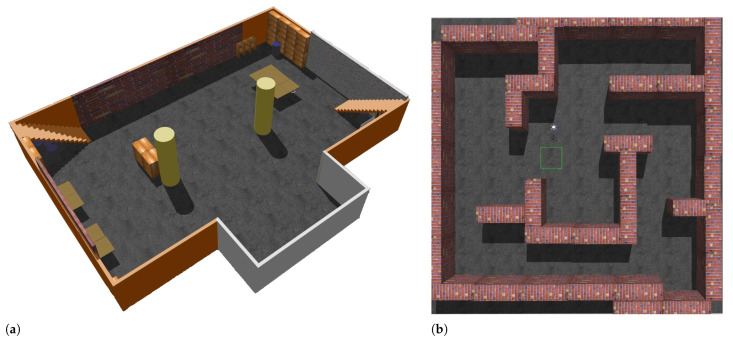
Two scenarios in Gazebo, (**a**) $20\times 12\times 3\;m^{3}$ indoor scenario. (**b**) $15\times 15\times 2\;m^{3}$ maze scenario, the green box depicts the initial position of the UAV.

**Figure 6 sensors-22-08429-f006:**
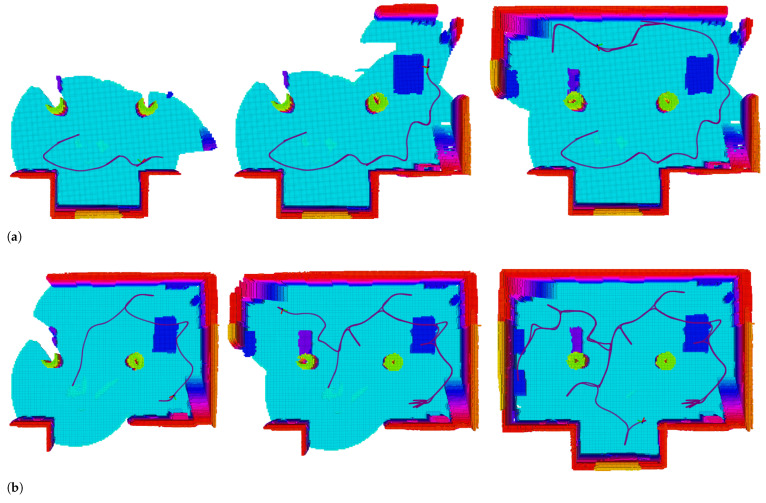
Exploration in indoor scenario, the three utility formations are compared, the color of the grid changes from blue to red as the height increases. (**a**) Exploartion using linear utility. (**b**) Exploartion using exponential utility. (**c**) Exploartion using normalized utility.

**Figure 7 sensors-22-08429-f007:**
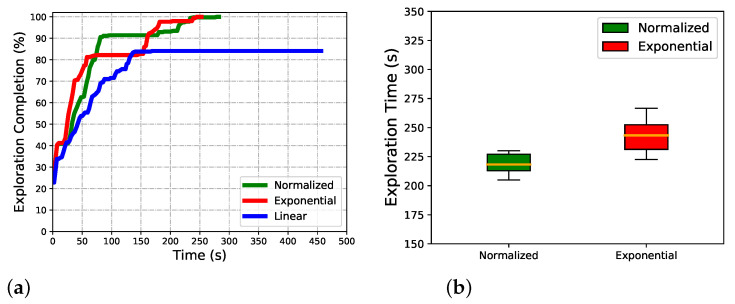
Algorithm comparsion in indoor scenario. (**a**) Exploring completion degree curve. (**b**) Total completion time.

**Figure 8 sensors-22-08429-f008:**
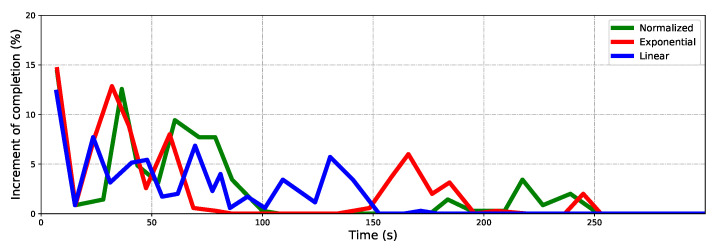
The increment of completion percent for three algorithms in the indoor scenario.

**Figure 9 sensors-22-08429-f009:**
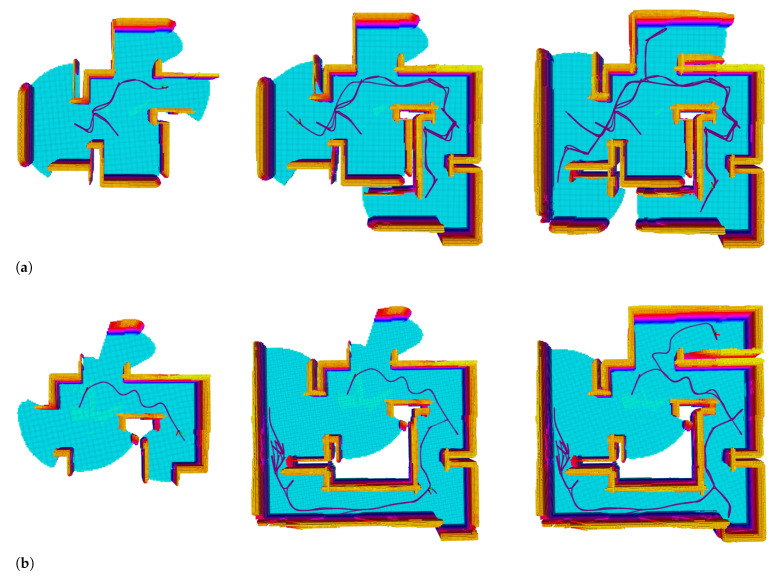
Exploration in maze scenario, the three utility formations are compared, the color of the grid changes from blue to red as the height increases. (**a**) Exploartion using linear utility. (**b**) Exploartion using exponential utility. (**c**) Exploartion using normalized utility.

**Figure 10 sensors-22-08429-f010:**
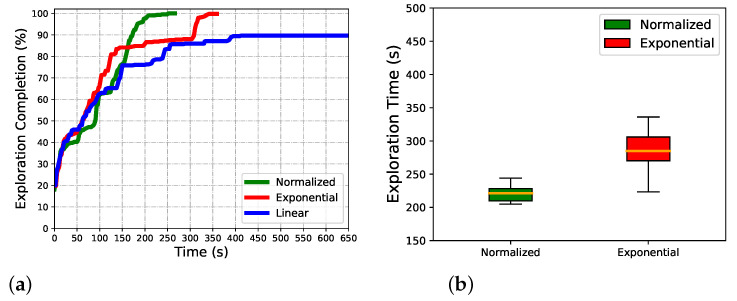
Algorithm comparison in maze scenario. (**a**) Exploring completion degree curve. (**b**) Total completion time.

**Figure 11 sensors-22-08429-f011:**
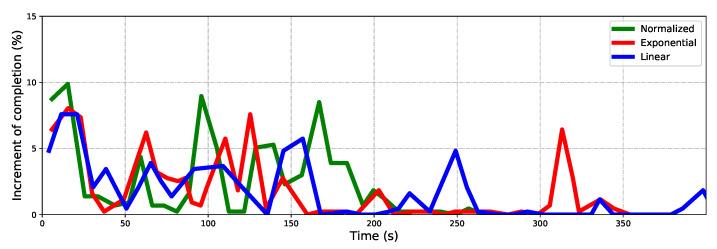
The increment of completion percentage for three algorithms in the maze scenario.

**Figure 12 sensors-22-08429-f012:**
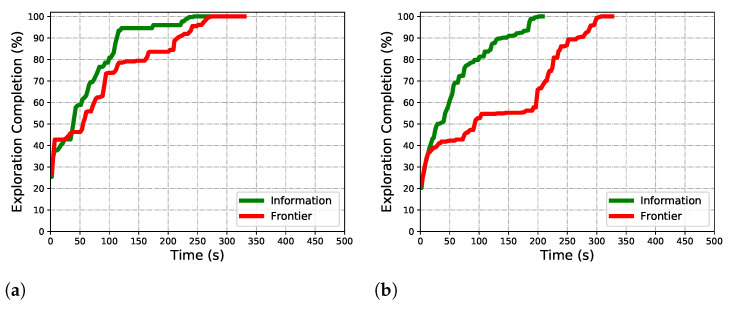
Algorithm comparison in maze scenario. (**a**) Exploring completion degree curve. (**b**) Total completion time.

**Table 1 sensors-22-08429-t001:** Parameters of the proposed planner.

Maximum segment length	$l_{max}$	1.2m
Sensor range	$d_{sensor}$	5m
Field of View	FOV	${\left[60,90\right]}^{\circ }$
Maximum velocity	$v_{max}$	1.5m/s
Maximum acceleration	$a_{max}$	$1\;m/s^{2}$
Maximum yaw velocity	$\phi_{max}$	1rad/s
Exponential parameter	λ	0.5
Linear parameter	λ	0.4

**Table 2 sensors-22-08429-t002:** Comparison of evaluation method.

	Indoor			Maze		
**Method**	**Evaluation (ms)**	**Total (s)**	**Trajectory (m)**	**Evaluation (ms)**	**Total (s)**	**Trajectory (m)**
Frontiers [13]	0.8±0.2	280.2±30.3	120.2±21.9	0.5±0.1	330.3±42.1	149.8±31.5
Information	3.1±2.7	220.5±21.8	115.1±20.2	2.1±1.7	225.8±35.2	127.2±20.6

## Data Availability

Not applicable.

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
