# Peer review of "Efficient Informative Path Planning via Normalized Utility in Unknown Environments Exploration"

_sensors, 2022, doi:10.3390/s22218429_

Round 1

Reviewer 1 Report

The paper presents a method for computing a drone trajectory so that the drone can observe an unknown environment. The method relies on extending a random tree and evaluating the branches that are generated in the tree. To evaluate the branches, it uses information gain normalized by the distance of the path. The method was evaluated in several scenarios and reaches competitive performance. However, the contribution should be clarified with respect to the previous work of Bircher et al. Indeed, a comparison against such work is desirable. 

At introduction, the difference with respect of the previous work is not clear 

In related work. There are several works that rely on the RRT for exploring an unknown region, even though they do not use a UAV, it is important to include them in the references. 

The problem is not well stated. The authors provide several concepts; however, the problem must be written in one sentence.  

Several references are omitted! such as for mapping and information gain computation and RRT formulation. 

Frontier method used in comparison is not referenced.

Reviewer 2 Report

The manuscript proposes a novel informative path planning method to realize efficient informative path planning in unknown exploration. The investigated topic is interesting and the manuscript has certain contributions. The paper can be refined by considering the following comments:

1. The literature review in the paper can be improved. First, when introducing the usage of robots to replace human work, more latest relevant research studies on using robots for efficiently package delivery in logistics can also be introduced such as ‘An integrated multi-population genetic algorithm for multi-vehicle task assignment in a drift field (2018)’, ‘Efficient routing for precedence-constrained package delivery for heterogeneous vehicles (2019)’ and ‘Integrated Task Assignment and Path Planning for Capacitated Multi-Agent Pickup and Delivery (2021)’. Second, more typical path planning approaches can be analyzed for robot path planning in unknown environments, such as the A* algorithm and Q-learning algorithm used in ‘Path Planning for Wheeled Mobile Robot in Partially Known Uneven Terrain’. These references are closely relevant to the path planning problem studied in the manuscript.

2. Some sentences are not complete or have gramma typos such as ‘A road map is topologies by RRT’, ‘An Efficient information richness judgment from posterior map entropy decline value’, ‘An informative path utility calculation method that normalizes information measurements dividing path distance’, and so on. Please further edit the manuscript.   

3. The objective function of the path planning problem is not clear. Please give more details in the problem description or formulate the objective function mathematically.  

4. Equation (1) can be refined as one row, and a comma or period is required at the end of equations (1), (2), (3), and (9).

5. More relevant popular path planning algorithms are suggested to be compared with the designed method in Section 5, and the computational running time of the algorithms is also needed for a fair comparison. 

Round 2

Reviewer 2 Report

My comments are dealt well with. Thanks for the authors' efforts to refine the paper.